# Enhanced Photoluminescence of Plasma-Treated Recycled Glass Particles

**DOI:** 10.3390/nano14131091

**Published:** 2024-06-25

**Authors:** Zdeněk Remeš, Oleg Babčenko, Vítězslav Jarý, Klára Beranová

**Affiliations:** FZU—Institute of Physics of the Czech Academy of Sciences, Na Slovance 1999/2, 182 00 Prague, Czech Republic; babcenko@fzu.cz (O.B.); jary@fzu.cz (V.J.); klara.beranova@fzu.cz (K.B.)

**Keywords:** recycling, silicon dioxide, glass, plasma, annealing, photoluminescence, Raman spectroscopy, X-ray photoelectron spectroscopy

## Abstract

Recycled soda-lime glass powder is a sustainable material that is also often considered a filler in cement-based composites. The changes in the surface properties of the glass particles due to the treatments were analyzed by X-ray photoelectron spectroscopy (XPS) and optical spectroscopy. We have found that there is a relatively high level of carbon contamination on the surface of the glass particles (around 30 at.%), so plasma technology and thermal annealing were tested for surface cleaning. Room temperature plasma treatment was not sufficient to remove the carbon contamination from the surface of the recycled glass particles. Instead, the room temperature plasma treatment of recycled soda-lime glass particles leads to a significant enhancement in their room temperature photoluminescence (PL) by increasing the intensity and accelerating the decay of the photoluminescence. The enhanced blue PL after room-temperature plasma treatment was attributed to the presence of carbon contamination on the glass surface and associated charge surface and interfacial defects and interfacial states. Therefore, we propose blue photoluminescence under UV LED as a fast and inexpensive method to indicate carbon contamination on the surface of glass particles.

## 1. Introduction

The conversion of glass waste into recycled building material can effectively reduce the stockpile of construction waste and minimize the extraction of natural resources [1].

Silicon dioxide-based glasses are able to retain their properties when recycled, and therefore, the most abundant soda-lime–silica glass is considered one of the most common recyclable materials [2]. Besides direct reuse in glass production, soda-lime glass powder is used in various industries, e.g., in glass wool production, as a binder for resins, as an additive for paints and adhesives, as an additive for brick and tile production, in the pyrotechnics industry, and in the cosmetics industry [3,4]. However, the variation in a glass waste composition, residual organic contamination, and additives significantly modifies the properties of different types of glass, affecting their possibility of recycling and creating a need to search for alternative ways of utilizing glass waste [2,5]. This is an especially important issue because SiO_2_ glass is not biodegradable and thus causes problems when landfilled. To reduce the environmental burden and decrease the use of natural sand, crushed or milled waste glass materials are very often considered as a promising filler in cement-based composites [6,7,8]. For added strength and lower costs, glass powder has been used in cement as an alternative to Portland cement or fly ash [9]. The review in [10] presents a comprehensive analysis of the utilization of clay brick powder in cement concrete and alkali-activated concrete. Concrete is one of the most widely and extensively used materials in the world. It is a mixture of various materials, aggregates (fine and coarse), cement, and water. Each of them is mixed in different amounts to achieve specific strengths. Recycled waste concrete fines are reused as cement or sand replacement, and the physical–chemical characteristics of the construction waste fines significantly impact the performance of the blended mortar [11].

Generally, recycled soda-lime glass is considered to be free of organic contaminants. However, in reality, the glass waste utilization process does not include sufficient steps leading to carbon-free milled glass waste powder [12], thus modifying glass powder surface properties and affecting its chemical reactivity. Therefore, efficient surface treatment technology is welcomed and needs to be studied [13,14].

Plasma contains highly reactive species, including electrons, ions, and free radicals [15]. When applied to clean glass, it can make the glass hydrophilic, increase its surface energy, and improve its wettability [16]. An improvement in the wettability of a glass surface plays a vital role in the bonding between the glass and other materials. After low-temperature plasma treatment, the wettability of the glass surface is expected to improve while the surface morphology should be almost unchanged. Molecular dynamics simulation results show that the active groups should be more easily adsorbed onto the glass surface in a free state and should interact strongly with water [17]. Therefore, the plasma treatment of glass powder has generated considerable interest in both basic and applied research. Recently, we modified the microcrystalline sand powder using a low-temperature, low-pressure plasma treatment and high-temperature annealing in different atmospheres [18,19] and observed that the low-temperature plasma treatment in hydrogen or oxygen atmospheres resulted in weak changes in the PL emission spectra. Significant changes in the PL emission spectra of SiO_2_ microparticles, attributed to changes in the defects in the glass particles, were observed only for annealing in hydrogen atmosphere at a temperature of 800 °C and above. Indeed, structural rearrangements of the glass matrix took place after the high-temperature process, affecting a wide range of glass properties [20].

In this paper, we focus on monitoring the surface properties of recycled soda-lime glass particles induced by high-temperature annealing in O_2_, H_2_, and N_2_ as well as plasma treatment. We show the crucial role of residual carbon contamination on the photoluminescence properties of room-temperature plasma-treated glass powder and propose photoluminescence measurements as a quick and economical alternative to XPS for the indication of carbon contamination in glass powders.

## 2. Materials and Methods

### 2.1. Recycled Glass Powder

A milled, commercially available recycled glass powder (Refaglass Trade, Ltd., Příbram, Czech Republic) was used in this study. The glass powder was made from a mixture of different colored shards (white, green, and brown). The soda-lime-based silica glass powder does not contain leaded or borosilicate glass and heavy metals, while the contents of the other chemical elements such as Fe, Cr, etc. varied slightly depending on the color ratio. The raw material was ground into a powder in a ball mill and passed through a rotary micro-sieve to remove detectable impurities.

### 2.2. Annealing and Plasma Treatment of Recycled Glass Powder

The as-received (i.e., taken from silos) glass powder was exposed to plasma in a large-area low-pressure system (AK 400, Roth&Rau, Hohenstein-Ernstthal, Germany) [21]. A radiofrequency (r.f.) plasma treatment was performed for 5 min in H_2_, N_2_, or O_2_ plasma at a pressure of 15 Pa and a temperature below 50 °C, marked as room temperature (RT), and repeated 4 times after glass powder mixing. High-temperature annealing was performed in H_2_, N_2_, or O_2_ at atmospheric pressure for a duration of 12 h in a cylindric furnace with a controllable atmosphere (Clasic CZ, Ltd., Revnice, Czech Republic). After all the complementary studies of glass powder annealing at a temperature of 500 °C, the r.f. plasma treatment of a powder performed at 500 °C was carried out. A total of 4 mg of powder was pressed onto a 10 × 10 mm^2^ Cu substrate in the form of disk pellets with a 3 mm diameter for optical measurements.

### 2.3. Scanning Electron Microscopy (SEM)

The size, shape, and surface morphology of the as-received glass powder particles were studied by scanning electron microscopy (SEM, e-Line system, Raith GmbH, Dortmund, Germany) at an electron acceleration voltage of 10 kV, employing an in-lens secondary electrons detector.

### 2.4. Fourier Transform Infrared Absorption and Raman Spectroscopy (FTIR)

Fourier transform infrared spectroscopy was measured using the dry-air, ventilated commercial Nicolet IS50 (FTIR) spectrometer and equipped with built-in attenuated total reflectance (ATR) and optional Raman accessory. ATR spectra were collected in the 100–1800 cm^−1^ spectral range with a 4 cm^−1^ spectral resolution using a heat source, solid state beamsplitter, diamond prism, and Peltier cooled DTGS detector. The absorbance spectra were spectrally corrected using the built-in software Advanced ATR Correction included in Omnic 9 software Raman measurements were collected in the 70–3500 cm^−1^ spectral range with a 2 cm^−1^ resolution using a YAG:Nd laser with excitation at 1064 nm (up to 0.5 W) and an InGaAs photodiode detector. FTIR spectrometer, all accessories and software were provided by Thermo Fisher Scientific, Ltd., Waltham, MA, USA.

### 2.5. X-ray Photoelectron Spectroscopy (XPS)

X-ray photoelectron (XPS) spectra were acquired using an AxisSUPRA photoelectron spectrometer (Kratos Analytical, Stretford, UK). The glass powders were spread in uniform layers to pieces of a Cu tape. An Al Kα monochromatized X-ray source, a hybrid lens mode, and charge compensation were used for all measurements. Resolution 80 was used when acquiring data for the purpose of estimating the atomic concentration of the elements. Additionally, resolution 10 was used for acquiring Si 2p; O 1s; C 1s; and in some cases, N 1s, regions for detailed analyses. The XPS spectra were shifted to lower binding energies due to charge compensation. To correct the spectra, the Si 2p3/2 peak was fixed at 103.7 eV. Moreover, because the measurements were taken through a long period of time, it was necessary to calibrate the intensities of the spectra (due to degrading of a detector). All spectra were normalized to intensities of Si 2p doublets. The atomic concentrations of the elements were calculated in the ESCApe, version 1.5.1.1051 (Kratos Analytical) (utilizing Kratos relative sensitivity factors and transmission functions determined for the AxisSUPRA photoelectron spectrometer in different settings of the analyzing system). Fitting was performed in the KolXPD, version 1.8.0 (build 68). The Shirley background was subtracted from all spectra. All spectral components were fitted by peaks or doublets of the Voigt profile. A ratio between the Lorentzian and Gaussian parts of the peaks were left free in various regions (0.3 at maximum).

### 2.6. Photoluminescence Spectroscopy

The steady-state PL was excited at room temperature using UV band pass filter #FBH360-10 transparent in 350–370 nm spectra range (Thorlabs, Newton, NJ, USA) and 1 mW UV LED #XSL-360-5E (Roither Lasertechnik, GmbH, Vienna, Austria) directly powered by a #3390 waveform generator (Keithley Instruments, subsidiary of Tektronix, Solon, OH, USA) [19]. Our setup measures the PL emission in the 375–750 nm spectral range using the #34-302 long-pass optical filter fully absorbing below 370 nm and fully transparent above 375 nm (Edmund Optics, Barrington, NJ, USA), a 300–800 nm monochromator H20VIS (Horiba, Kyoto, Japan), a red-sensitive photomultiplier (PMT) #XP2203B (Photonis, Orsay, France), a low-noise current preamplifier #5182 with 10^5^ V/A transimpedance (AMETEK, Berwyn, PA, USA), and a 100 kHz #SR830 lock-in amplifier (Stanford Research Systems, Sunnyvale, CA, USA) TTL referenced to the LED frequency. The setup is also suitable for time-resolved PL measurements, with the time resolution in the order of 10 ns using the method based on the phase shift between the sinusoidal excitation and emission. Prior to the phase shift measurements, the lock-in amplifier was set to zero phase at the UV LED wavelength [19]. The setup was spectrally calibrated with an #63358 halogen lamp (Oriel Instruments, subsidiary of Newport Corporation, Stratford, CT, USA)).

The time-resolved PL was also measured at a selected wavelength directly using a custom-made spectrofluorometer 5000 M (Horiba Jobin Yvon, Wildwood, MA, USA) using a nanosecond nanoLED ns pulsed light (Horiba Scientific, Piscataway, NJ, USA) as the excitation sources in time-correlated single-photon counting mode (TCSPC). The detection part of the setup involved a single-grating monochromator and a photon-counting detector TBX-04 (Hamamatsu, Photonics, Hamamatsu City, Shizuoka, Japan). The convolution procedure was applied to the photoluminescence decay curves to determine the true decay times (SpectraSolve software package 3.01 PRO, Ames Photonics, Fort Worth, TX, USA).

## 3. Results

### 3.1. Scanning Electron Microscopy

The typical electron microscopy image of the as-received micro-milled waste glass powder is shown in Figure 1. According to the SEM observations, the used powder consists of large (tens of micrometers), irregular, chipped pieces decorated with small fragments (tens of nanometers). It needs to be added that the observed particles have cracks and fractures that increase the particle’s surface (as the interface between particle volume and environment). Such a mixture of larger and smaller particles requires a universal technique able to analyze particle properties regardless of size.

### 3.2. Infrared Absorption Spectra

Figure 2 shows that the infrared absorption spectra of the studied powder are dominated by three absorption bands corresponding to Si-O-Si asymmetric stretching vibration (1050 cm^−1^), Si-O-Si symmetric stretching (790 cm^−1^), and Si-O-Si rocking (470 cm^−1^) [22]. These bands are broad due to the amorphous nature of the glass samples. Most interesting is the fact that the vibrations do not significantly change during the plasma treatment or annealing. We conclude that no short-range bond rearrangement or recrystallization occurred in our samples following high-temperature and plasma treatment. Moreover, no CH-related stretching vibrations (at around 3000 cm^−1^) were detected for the as-received [12] or plasma-treated powders (not shown in Figure 2a).

### 3.3. IR Raman Spectra

Figure 3 shows that the IR Raman spectra are dominated by the inelastic scattering of hydroxyl group OH. The OH group is present in glass powder even after plasma treatment at 500 °C, and it was reduced significantly only after high-temperature annealing in oxygen atmosphere at 700 °C. However, 700 °C caused melting of the glass powder (due to soda additives) [13,20].

### 3.4. X-ray Photoelectron Spectra (XPS)

The XPS measurements reveal a relatively high concentration of carbon at the surface of the recycled glass powder. Such an amount of carbon contamination is common for Si-glass materials exposed to the natural environment [23]. As can be seen from Table 1, carbon contamination remains in the samples even after annealing at 500 °C, as well as after plasma treatment. Contrary to the results from IR absorption and Raman spectroscopy, carbon contamination was detected only with the surface sensitive XPS method. We deduce from this that organic residues are adsorbed in the form of an ultra-thin layer (in nm order) on the glass particles’ surface.

Following the procedure described by Smith et al. [24] and approximating the carbon-contamination layer as a homogeneous and uniformly thick film on the surface of the glass particles, we estimated that the thickness of the carbon-contamination layer ranges from 0.4 (11 at% of C) to 1.7 nm (39 at% of C). The detection limit of XPS in a SiO_2_-based material ranges from roughly 4 nm for low-energy photoelectrons (300 eV) to 10 nm for high-energy photoelectrons (1200 eV), considering that it is a three-times multiple of the inelastic mean free path (1.25 and 3.35 nm, respectively) [25,26]. In comparison to the detection limit of XPS, the carbon-contamination layer contributes significantly to the overall XPS signal and should not be omitted from analysis. Therefore, we attempted to include carbon contaminants to the decompositions of the photoelectron spectra.

Figure 4 shows the O 1s and C 1s XPS spectra decomposed to different bonding states. The C 1s region (Figure 4a) was fitted by four peaks corresponding to carbon bonded to C-C aliphatic carbon (285.9 eV), C-O, C-OH or C-N (287.1 eV), C=O (289.5 eV), and HO-C=O (290.8 eV) [27,28]. Unusually, a high binding energy of C-C aliphatic carbon is probably the result of calibrating the binding energies of the XPS spectra to the Si 2p doublet from the silicon network [29].

The O 1s region (Figure 4b) was fitted by a total of six peaks. Three peaks located at approximately 531.7, 533.5, and 534.6 eV originated from the silicon network and can be attributed to non-bridging oxygen (NBO: Si-O-M, where M represents alkali or alkaline earth metal ions), bridging oxygen (BO: Si-O-Si), and the hydrous species Si-OH/H_2_O, respectively [30]. Three additional peaks representing carbon–oxygen contamination were included in the fit: C-O(H), COOH, and C=O. These components were positioned at 1.6, 2.8, and 4.2 eV lower binding energies relative to the peak of H_2_O in accordance with [31,32]. Because the O 1s spectrum did not have sufficient resolution to fit all the overlapping components without any restraints, we fixed the areas of the carbon–oxygen peaks to those derived from C 1s region decomposition. In these constraints, we considered that one carbon atom from the C 1s C-O(H) would contribute one oxygen atom to the O 1s C-O(H) (C-N was neglected), one C 1s C=O would contribute one O 1s C=O, and one C 1s COOH would contribute two oxygen atoms to the O 1s COOH. The structure at around 538 eV originates from Na KLL Auger electrons [33].

The amount of carbon was partly reduced after high-temperature annealing combined with O_2_ or H_2_ plasma treatment. On the other hand, room-temperature plasma treatment has a very low effect on C concentration. The relatively high concentration of Na and Ca in the sample is related to the soda-lime nature of the recycled glass. A low concentration of N was detectable after N_2_ plasma treatment. It should be noted that the room-temperature N_2_ plasma treatment was more efficient in binding N on glass than the high temperature plasma treatment, probably because of post-treatment glass powder cooling from 500 °C in the system chamber.

Figure 5 demonstrates the ratios of different bonding states at the sample surface as derived from XPS. The black squares show the contribution of non-bridging oxygen to the silicon network in the surface area (ranges from 18 to 28%). We can see that NBO concentration in the silicon network is mostly constant, slightly increasing at a higher temperature. This indicates that treatment in plasma or gases influences only the top-most surface layer and does not alter the composition of the silicon network. Green triangles represent the relative contribution of carbon–oxygen contamination to the O 1s spectra. We can see that annealing at 500 °C results in a significant decrement of carbon–oxygen species, which agrees with a decrease in the overall carbon concentration (See Table 1). The red circles represent the ratio of H_2_O and Si-OH species to oxygen signal from the silicon network. We can see that the sample O_2_-500 °C (low concentration of carbon, high oxygen and sodium) shows most affinity to water absorption.

### 3.5. Photoluminescence Spectra

Figure 6 shows the PL spectra of glass powder oxygen processing. Similar results were observed also after H_2_ and N_2_ processing. The PL spectra of as-received and annealed-at-500 °C (no plasma) samples were added for comparison. The increase in blue PL after room-temperature plasma treatment is more than an order of magnitude as compared with the as-received powder. We attribute the observed increase in blue PL intensity after RT plasma treatment to the organic contaminants that resulted in a PL active surface and interface defect under the condition of plasma exposure. We have also found that the PL intensity of plasma-treated recycled glass powder deteriorated when the plasma treatment had been realized at elevated temperatures, i.e., when carbon contaminants should be removed by active oxygen or hydrogen. It agrees with the XPS observation that demonstrates a significant reduction in carbon content in the glass powder after annealing or plasma treatment at high temperatures (up to three times).

Figure 7a shows the enhanced PL spectrum and the PL decay of plasma-treated glass powder at room temperature. The PL decay is very fast, especially in a blue spectral range, with the PL mean time decay about 1 ns. The time-resolved PL decay at the selected wavelength (450 nm) shown in Figure 7b confirms the fast PL decay in the order of 1 ns, but they also reveal two slow PL decay components with a mean decay time of about 8 ns and 3 µs, see Table 2.

## 4. Discussion

XPS shows that carbon impurities (both from the total carbon concentration and from the oxygen–carbon impurities) decrease after high-temperature annealing. The NBO/(NBO + BO) ratio is approximately constant, increasing slightly at higher temperatures, which may be because we have removed some of this carbon. Note that the standard error is about 10%. This is the ratio of two values with a huge standard error. But, a large part of this error is the systematic error, which, although relatively large, is the same for all spectra due to the procedure and the way the data were handled. This is why the data from each sample can be compared reliably.

Vibrational spectroscopies are one of the most powerful, fast, and reliable methods for materials’ structure identification and characterization [34]. Previous studies suggest that thermal treatment at an elevated temperature can cause changes in the structure of the glass, which can be detected by changes in the Raman and IR absorption spectra [35]. However, no significant changes were observed in our IR absorption and Raman spectra.

Previous studies have suggested that the photoluminescence of soda-lime glass powder can be manipulated by adding certain elements or compounds [36,37]. However, the specific changes would depend on the composition of the soda-lime glass and the conditions of the thermal treatment. While specific studies on soda-lime glass powder are not readily available, we have observed that plasma treatment at elevated temperatures and high-temperature annealing (not exceeding 500 °C) resulted in changes in the XPS and PL spectra. The most significant changes were observed after plasma treatment at room temperature in the blue PL emission spectra. The enhanced blue PL emission of plasma-treated glass powder is very similar regardless of the type of plasma. Therefore, we suggest that the enhanced blue PL after room-temperature plasma treatment is related to the presence of carbon contamination on the glass surface and related charge interface defects and interface states [38]. At elevated temperatures, the carbon content is reduced and changes its state to graphitic (mainly C-C bonds in the C 1s XPS spectra in Figure 3), and therefore, the blue PL “disappears”.

## 5. Conclusions

There are only a few studies on the surface properties of glass waste particles after plasma and thermal cleaning, and the current study focuses on this scientific problem. In particular, it is of great practical importance to study the effect of low-temperature plasma treatment on the surface properties of glass particles analyzed by surface sensitive techniques. The recycled glass powder was modified by plasma and high-temperature annealing in oxygen, hydrogen, and nitrogen atmospheres. The change in surface and optical properties of recycled glass particles was investigated by XPS, IR absorption, and Raman and photoluminescence spectroscopy. We explain the enhanced blue PL after room-temperature plasma treatment based on the modification of carbon contaminants detected on the glass surface by the XPS and associated interfacial defects and interfacial states. The observed results demonstrate the usefulness of plasma treatment for the surface modification/decontamination of recycled glass particles as well as the high potential of photoluminescence in detecting ultrathin layers of organic contaminants on the glass particle surface that allow fast and inexpensive surface control.

## Figures and Tables

**Figure 1 nanomaterials-14-01091-f001:**
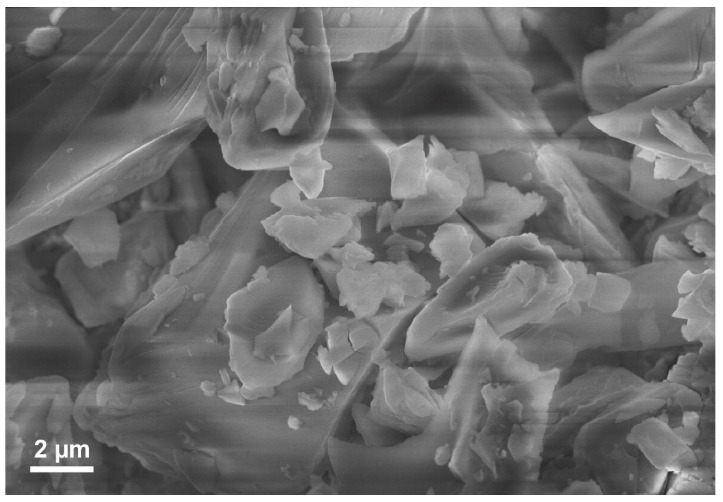
SEM image of the as-received micro-milled waste glass powder.

**Figure 2 nanomaterials-14-01091-f002:**
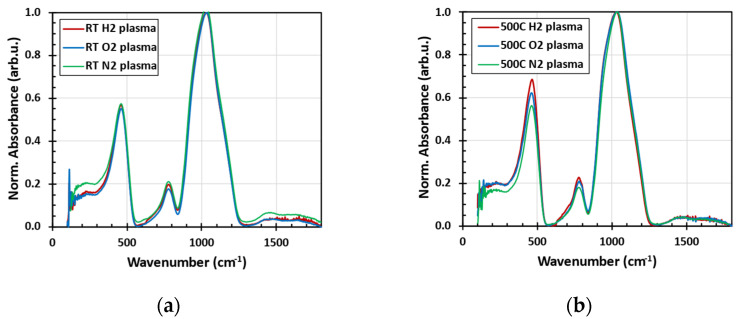
The normalized FTIR absorbance spectra measured in an FTIR spectrometer with an ATR accessory after low-pressure plasma treatment at room temperature (**a**) and at 500 °C (**b**).

**Figure 3 nanomaterials-14-01091-f003:**
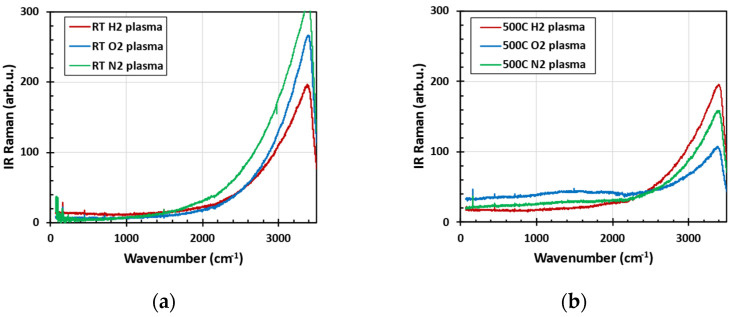
IR Raman spectra measured with an FTIR spectrometer with 1064 nm laser excitation after low-pressure plasma treatment at room temperature (**a**) and at 500 °C (**b**).

**Figure 4 nanomaterials-14-01091-f004:**
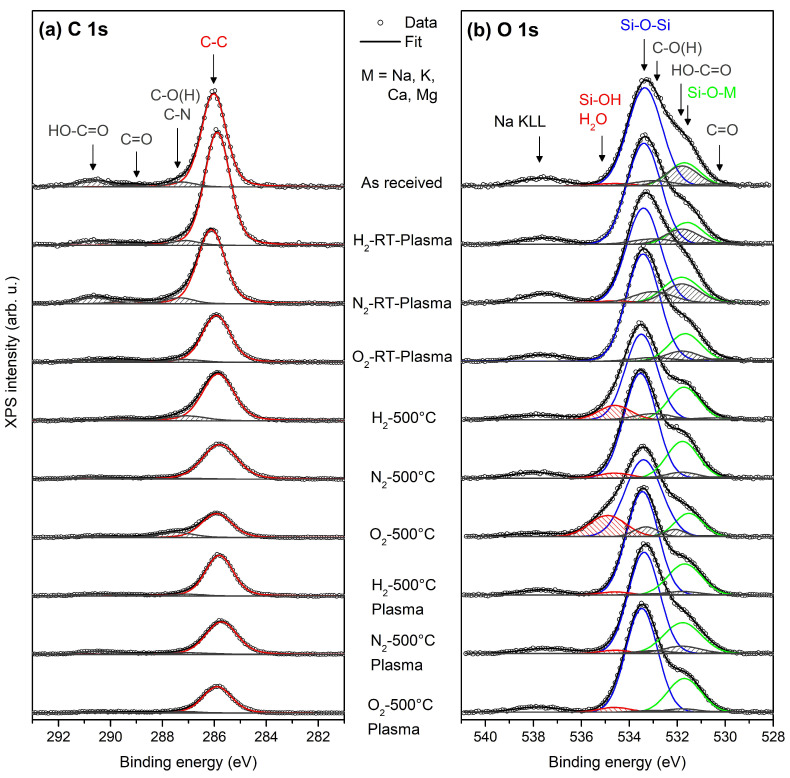
C 1s (**a**), and O 1s (**b**) XPS spectra (from top to bottom) for the “as-received” sample, with the samples treated in H_2_, N_2_, and O_2_ plasma at room temperature; annealed in H_2_, N_2_, and O_2_ atmosphere at 500 °C; and annealed in H_2_, N_2_, and O_2_ plasma at 500 °C. Dots represent the measured data, black lines are the results of fits, and colored lines are the fitting components. The peaks shaded by grey originate from carbon–oxygen contamination.

**Figure 5 nanomaterials-14-01091-f005:**
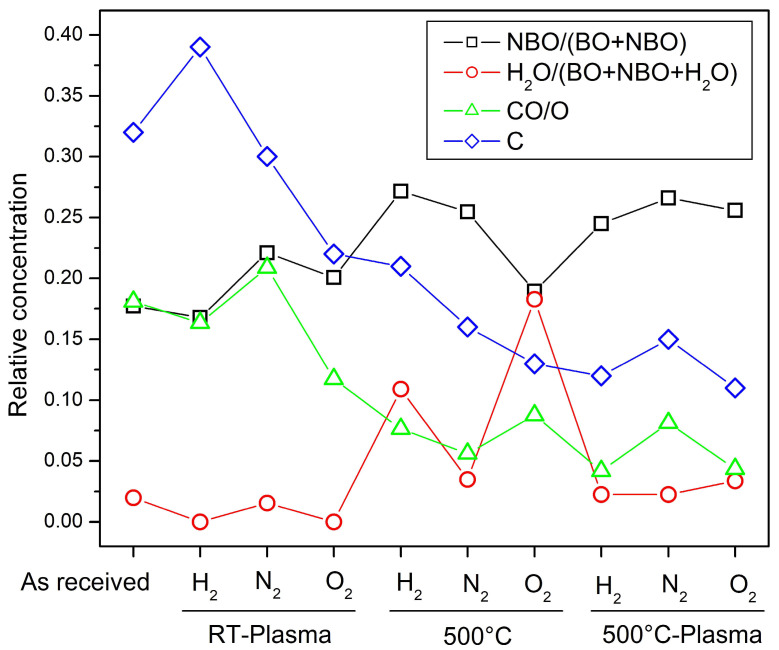
Ratio of selected components derived from fitting XPS spectra. We use the following labels for simplification: BO: Si-O-Si bridging oxygen, NBO: Si-O-M non-bridging oxygen, H_2_O: oxygen from hydrous species bound to silicon (H_2_O, Si-OH), CO: carbon–oxygen contaminations (C-O(H), C=O, COOH), C: atomic concentration of carbon normalized to 1.

**Figure 6 nanomaterials-14-01091-f006:**
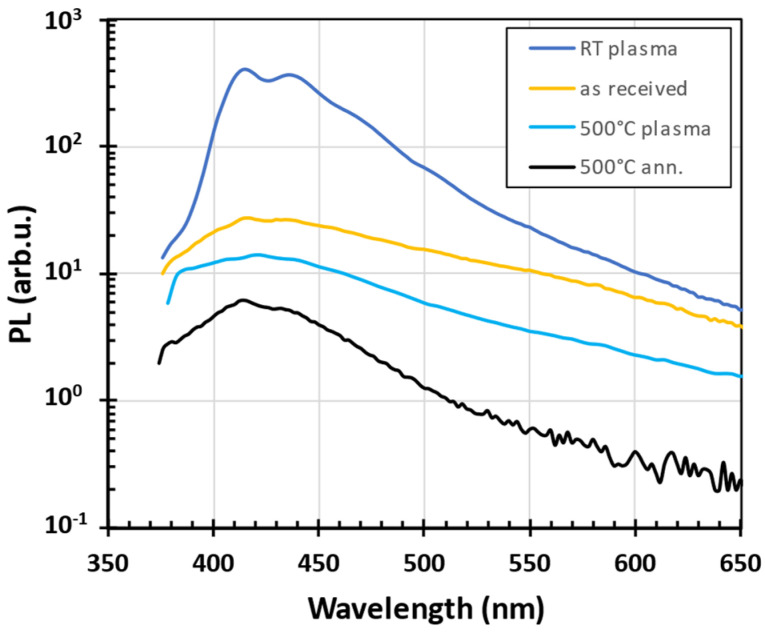
PL spectra of treated glass powder oxygen plasma. “RT plasma”, resp. “500 °C plasma” represents the PL spectra measured after room temperature, resp. 500 °C O_2_ plasma treatment, the PL spectra of as-received and annealed-at-500 °C-in-O_2_-atmosphere (no plasma) glass powder were added for comparison.

**Figure 7 nanomaterials-14-01091-f007:**
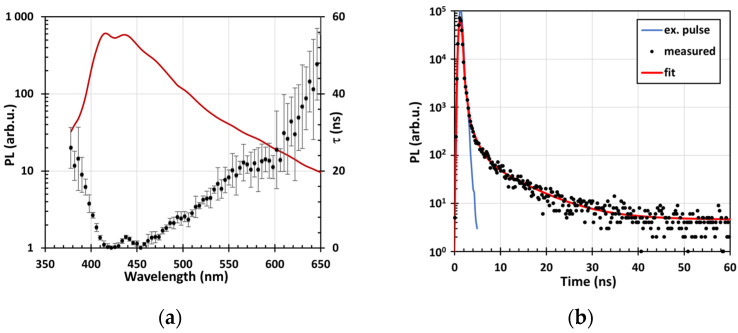
PL spectra (red curve, left axis) of room-temperature H_2_ plasma-treated glass powder using sinusoidal UV LED excitation at a frequency of 100 kHz and the spectrally resolved mean time of PL decay (black points with error bars, right axis) evaluated from the phase shift between the excitation and emission (**a**). The curve with error bars corresponds to the spectrally resolved mean PL decay time τ calculated from the phase shift between the excitation and the emission spectra. The time-resolved PL emission measured by TCSPC at 450 nm is shown in (**b**). The instrumental response shape measured at the excitation pulse wavelength (blue curve) and fitted function (red curve) were added in (**b**) for comparison. All measurements were performed at RT.

**Table 1 nanomaterials-14-01091-t001:** Recycled glass powder atomic % concentration evaluated from XPS.

Sample	O	Si	C	Na	Ca	N	Al	Mg
as received	40	17	32	7	2	<1	1	1
H_2_-RT-Plasma	36	17	39	6	2	<1	1	1
N_2_-RT-Plasma	40	17	30	8	2	2	1	1
O_2_-RT-Plasma	45	20	22	8	2	<1	1	1
H_2_-500 °C	46	19	21	9	2	<1	1	1
N_2_-500 °C	48	20	16	11	3	<1	1	1
O_2_-500 °C	51	20	13	10	3	<1	1	1
H_2_-500 °C-Plasma	50	25	12	8	2	<1	1	1
N_2_-500 °C-Plasma	49	20	15	11	2	1	1	1
O_2_-500 °C-Plasma	50	25	11	8	2	<1	1	1

**Table 2 nanomaterials-14-01091-t002:** The time-resolved PL decay curves $I\left(t\right)=A_{0}+\sum_{i=1}^{n}A_{i}e^{-\frac{t}{\tau_{i}}}$ of room-temperature plasma-treated recycled glass powder approximated by the sum of exponentials convolved with the instrumental response function, where *I*(*t*), *A_i_*, and *τ_i_* stand for the intensity as a function of time *t* (expressed as the number of photons detected in the time interval), amplitude (in arbitrary units), and decay time (in ns) of the *i*th component, respectively. *A*_0_ represents the background.

RT Plasma	*A*_0_ (a.u.)	*A*_1_ (a.u.)	*A*_2_ (a.u.)	*A*_3_ (a.u.)	*τ*_1_ (ns)	*τ*_2_ (ns)	*τ*_3_ (ns)
H_2_	0.117	1530	130	5	1.4	7.7	2900
O_2_	0.197	2580	130	5	1.3	8.5	3060
N_2_	0.211	6880	340	9	1.1	7.7	3300

## Data Availability

The data are available in a public ASEP repository of the Czech Academy of Sciences (https://asep.lib.cas.cz/arl-cav/cs/detail-cav_un_epca-0586898-Dataset-for-Enhanced-photoluminescence-of-plasma-treated-recycled-glass-particles/), accessed on 29 May 2024).

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
