# Peer review of "Enhanced Photoluminescence of Plasma-Treated Recycled Glass Particles"

_nanomaterials, 2024, doi:10.3390/nano14131091_

Round 1

Reviewer 1 Report

Comments and Suggestions for Authors

This article investigates the Enhanced photoluminescence of plasma treated recycled glass particles. Overall, the topic is significant, and the research findings contribute to the recycling and resource utilization of waste materials. Consequently, this paper could be considered for publication after revisions. The author should take the following suggestions into account for the revised manuscript.

1. Please emphasize the novelty of this study in the abstract, highlighting how it differs from previous research. Additionally, quantitative data should be included to provide readers with a clear understanding of the key data points.

2. The introduction is somewhat weak; the author should supplement it with an overview and analysis of relevant literature from both domestic and international sources, underscoring the deficiencies and shortcomings of prior studies, and ultimately clarifying the objectives and significance of this research.

3. At the beginning of the introduction, the author should provide a description of the resource utilization of waste materials. The following two articles may be helpful for this paper. (a) Reusing waste clay brick powder for low-carbon cement concrete and alkali-activated concrete: A critical review; (b) Micro-macro characterizations of mortar containing construction waste fines as replacement of cement and sand: A comparative study.

4. The discussion and conclusion sections are somewhat weak; it is recommended that more description and analysis be added. Particularly in the conclusion section, the author should list the main findings in 4-6 points.

Comments on the Quality of English Language

 Moderate editing of English language required

Author Response

  1. Please emphasize the novelty of this study in the abstract, highlighting how it differs from previous research. Additionally, quantitative data should be included to provide readers with a clear understanding of the key data points.

We point out the novelty of this work by the statement that indicate deficiency of knowledge related to surface of glass particles "Even though for powder materials properties of surface are crucial in the waste glass reuse the particles surface is not always examined enough.", also "Room temperature plasma treatment was not sufficient to remove carbon contamination. Therefore, we propose room temperature photoluminescence spectroscopy as a fast and direct method to indicate carbon contamination on the surface of in glass nanoparticles"....

The quantitative statement "However, there is a relatively high level of carbon contamination on the surface of glass nano-particles (around 30 at.%), so plasma technology and thermal annealing were tested for surface cleaning of recycled glass waste-based nanoparticles." has been added into the Abstract.

  1. The introduction is somewhat weak; the author should supplement it with an overview and analysis of relevant literature from both domestic and international sources, underscoring the deficiencies and shortcomings of prior studies, and ultimately clarifying the objectives and significance of this research.

The "Introduction" was substantially expanded with new quotes added. The “Introduction” now cites 19 literature sources and summarizes the role of plasma processing in the treatment of recycled waste.

  1. At the beginning of the introduction, the author should provide a description of the resource utilization of waste materials. The following two articles may be helpful for this paper. (a) Reusing waste clay brick powder for low-carbon cement concrete and alkali-activated concrete: A critical review; (b) Micro-macro characterizations of mortar containing construction waste fines as replacement of cement and sand: A comparative study

The recommended citations were added into the Introduction

Mohammadi Golafshani, E.; Kashani, A.; Behnood, A.; Kim, T. Modeling the Chloride Migration of Recycled Aggregate Concrete Using Ensemble Learners for Sustainable Building Construction. Journal of Cleaner Production 2023, 407, 136968, doi:10.1016/j.jclepro.2023.136968.

 Wu, H.; Gao, J.; Liu, C.; Guo, Z.; Luo, X. Reusing Waste Clay Brick Powder for Low-Carbon Cement Concrete and Alka-li-Activated Concrete: A Critical Review. Journal of Cleaner Production 2024, 449, 141755, doi:10.1016/j.jclepro.2024.141755.

  1. The discussion and conclusion sections are somewhat weak; it is recommended that more description and analysis be added. Particularly in the conclusion section, the author should list the main findings in 4-6 points.

The “Discussion” and “Conclusion” sections has been rewritten and extended. New citations were added.

Reviewer 2 Report

Comments and Suggestions for Authors

SUMMARY

The article submitted for review is relevant to modern science. It addresses the issues of improved photoluminescence of plasma-treated recycled glass particles. The relevance and importance of this study is due to the fact that recycled soda-lime glass is an environmentally friendly material that is also often used as a filler in cement-based composites. At the same time, there is a problem expressed in certain restrictions on the effective use of glass waste. Therefore, plasma technology and thermal firing were tested to treat the surface of glass waste-based microparticles.

The authors solved the problem using the methods of X-ray photoelectron spectroscopy and optical spectroscopy. The results of the study show the potential of plasma treatment as an effective method for modifying the surface properties of waste glass-based particles.

The authors conducted a very important study, which has both scientific novelty and practical significance. The reviewer highly appreciates the work of the authors, but believes that a number of serious corrections should be made. At the moment, the article does not yet meet the requirements of the scientific journal “Nanomaterials” and needs extensive revision.

COMMENTS

1.      The first remark I would like to note is the insufficient substantiation of the stated topic in the subject of the Nanomaterials journal. The authors report that they worked at the microparticle level. It should be understood that the Nanomaterials journal, as a rule, publishes research related to nanoparticles. Therefore, the authors need to add a clearer and more detailed justification for the correspondence of the subject of the article to the subject of the Nanomaterials journal.

2.      Authors then need to revise their abstract. There is no scientific problem in the abstract. The authors talk about the limitations of using waste glass, but do not say what scientific deficiencies exist. It should be said that there are gaps in knowledge of the surface properties of microparticles based on glass waste after plasma thermal firing technology and that the study eliminates precisely this scientific problem.

3.      The authors report in the abstract that they have obtained important results related to the potential of plasma processing of waste glass particles. It should be shown how much better this technology is than its analogues. By what percentage does this method show better parameters than previously known or alternative ones? The result must be expressed quantitatively, otherwise the abstract does not reflect the significance of the article.

4.      The “Introduction” section is very small; only 13 literature sources were analyzed. For such a study on such a pressing topic, it is recommended to analyze at least 30-35 sources of literature. The current state of the issue should be reflected more clearly and extensively.

5.      The section “Materials and Methods” is written in continuous text. This section would benefit from flowcharts of the study program, clearly indicating study inputs and outputs. There are also not enough tables describing the properties of the materials used, as well as methodology in the form of a step-by-step algorithm. In its current form, the “Materials and Methods” section is very difficult to understand.

6.      The “Discussion” section is very small and contains references to only one source of literature. The authors need to make a more detailed comparison of the results with those of other authors. It must be proven that the scientific novelty and practical significance of the authors are high and correspond to the scientific level of the journal.

7.      Noteworthy is the absence of photographs in the article. I would like to add some illustrative material. The article looks uninformative and has the risk of not arousing increased interest from readers. Although, of course, the idea is very interesting.

8.      Conclusions need to be systematized. They need to be numbered and arranged as follows: scientific novelty, practical significance, prospects for the development of research in the future and recommendations for the real industry.

9.      The list of references does not reflect the current state of the issue. A total of 18 literature sources are provided, some of which are older than the last 5 years. The authors need to do a good job of reviewing the literature on the research topic.

In general, taking into account all the comments made, the article does not yet look ready for publication in the Nanomaterials journal; serious revision is needed. The reviewer invites the authors to make all changes according to the comments made, and after correcting the article, he would like to look at it again. The general conclusion is major revisions.

Author Response

  1. The first remark I would like to note is the insufficient substantiation of the stated topic in the subject of the Nanomaterials journal. The authors report that they worked at the microparticle level. It should be understood that the Nanomaterials journal, as a rule, publishes research related to nanoparticles. Therefore, the authors need to add a clearer and more detailed justification for the correspondence of the subject of the article to the subject of the Nanomaterials journal.

To justify for the correspondence of the subject of the article to the subject of the Nanomaterials journal we have included into the Abstract the statement "Room temperature plasma treatment was not sufficient to remove carbon contamination from the surface of recycled glass nanoparticles. Therefore, we propose room temperature photoluminescence spectroscopy as a fast and direct method to indicate carbon contamination on the surface of in glass nanoparticles.". We have also added SEM image that shows that the powder is a mixture of microparticles as well as nanoparticles. However, since the revised paper is focused on the surface-related XPS and PL, the nanoparticles dominate the measured data because of their large surface area relative to the microparticles.

  1. Authors then need to revise their abstract. There is no scientific problem in the abstract. The authors talk about the limitations of using waste glass, but do not say what scientific deficiencies exist. It should be said that there are gaps in knowledge of the surface properties of microparticles based on glass waste after plasma thermal firing technology and that the study eliminates precisely this scientific problem.

The abstract has been rewritten to emphasize the problem of detecting carbon contamination on the surface of glass waste nanoparticles by XPS and PL spectroscopy.

  1. The authors report in the abstract that they have obtained important results related to the potential of plasma processing of waste glass particles. It should be shown how much better this technology is than its analogues. By what percentage does this method show better parameters than previously known or alternative ones? The result must be expressed quantitatively, otherwise the abstract does not reflect the significance of the article.

The abstract and conclusion were rewritten to be focused on plasma cleaning and detecting carbon contamination on the surface of glass waste nanoparticles by XPS and PL spectroscopy.

  1. The “Introduction” section is very small; only 13 literature sources were analyzed. For such a study on such a pressing topic, it is recommended to analyze at least 30- 35 sources of literature. The current state of the issue should be reflected more clearly and extensively.

The "Introduction" was substantially expanded with new quotes added. The “Introduction” now cites 20 literature sources and summarizes the role of plasma processing in the treatment of recycled waste. The revised paper cites in total 38 references.

  1. The section “Materials and Methods” is written in continuous text. This section would benefit from flowcharts of the study program, clearly indicating study inputs and outputs. There are also not enough tables describing the properties of the materials used, as well as methodology in the form of a step-by-step algorithm. In its current form, the “Materials and Methods” section is very difficult to understand.

The section “Materials and Methods” was spilt into 6 sub-sections: 2.1 Recycled glass powder; 2.2 Annealing and plasma treatment of recycled glass powder; 2.3 Scanning electron microscopy (SEM); 2.4 Fourier transform Infrared absorption and Raman spectroscopy (FTIR); 2.5 X-ray photoelectron spectroscopy (XPS); 2.6  Photoluminescence spectroscopy

  1. The “Discussion” section is very small and contains references to only one source of literature. The authors need to make a more detailed comparison of the results with those of other authors. It must be proven that the scientific novelty and practical significance of the authors are high and correspond to the scientific level of the journal.

The “Discussion” section has been rewritten and extended. New citations were added.

  1. Noteworthy is the absence of photographs in the article. I would like to add some illustrative material. The article looks uninformative and has the risk of not arousing increased interest from readers. Although, of course, the idea is very interesting.

The Figure 1. SEM image of the as received recycled glass powder was added.

  1. Conclusions need to be systematized. They need to be numbered and arranged as follows: scientific novelty, practical significance, prospects for the development of research in the future and recommendations for the real industry.

The conclusions were rewritten as suggested by referee.

  1. The list of references does not reflect the current state of the issue. A total of 18 literature sources are provided, some of which are older than the last 5 years. The authors need to do a good job of reviewing the literature on the research topic. In general, taking into account all the comments made, the article does not yet look ready for publication in the Nanomaterials journal; serious revision is needed. The reviewer invites the authors to make all changes according to the comments made, and after correcting the article, he would like to look at it again. The general conclusion is major revisions.

The authors made suggested changes, in particular The Abstract, Introduction and Conclusion sections were rewritten and new references were added to reflect the current state of the issue.

Round 2

Reviewer 2 Report

Comments and Suggestions for Authors

The authors clearly responded to all the reviewer's comments. The reviewer had no more questions. In its current form, the manuscript is suitable for publication in a journal.